# Global Composition of the Bacteriophage Community in Honey Bees

Taylor J. Busby,[a] Craig R. Miller,[b,c] Nancy A. Moran,[d] James T. Van Leuven[b,c]

aGlobal Disease Biology, University of California, Davis, Davis, California, USA

bDepartment of Biological Sciences, University of Idaho, Moscow, Idaho, USA

cInstitute for Modeling Collaboration and Innovation, University of Idaho, Moscow, Idaho, USA

dDepartment of Integrative Biology, University of Texas at Austin, Austin, Texas, USA

**ABSTRACT** The microbial communities in animal digestive systems are critical for host development and health. They stimulate the immune system during development, synthesize important chemical compounds like hormones, aid in digestion, competitively exclude pathogens, etc. Compared to the bacterial and fungal components of the microbiome, we know little about the temporal and spatial dynamics of bacteriophage communities in animal digestive systems. Recently, the bacteriophages of the honey bee gut were characterized in two European bee populations. Most of the bacteriophages described in these two reports were novel, harbored many metabolic genes in their genomes, and had a community structure that suggests coevolution with their bacterial hosts. To describe the conservation of bacteriophages in bees and begin to understand their role in the bee microbiome, we sequenced the virome of *Apis mellifera* from Austin, TX, and compared bacteriophage compositions among three locations around the world. We found that most bacteriophages from Austin are novel, sharing no sequence similarity with anything in public repositories. However, many bacteriophages are shared among the three bee viromes, indicating specialization of bacteriophages in the bee gut. Our study, along with the two previous bee virome studies, shows that the bee gut bacteriophage community is simple compared to that of many animals, consisting of several hundred types of bacteriophages that primarily infect four of the dominant bacterial phylotypes in the bee gut.

**IMPORTANCE** Viruses that infect bacteria (bacteriophages) are abundant in the microbial communities that live on and in plants and animals. However, our knowledge of the structure, dynamics, and function of these viral communities lags far behind our knowledge of their bacterial hosts. We sequenced the first bacteriophage community of honey bees from the United States and compared the U.S. honey bee bacteriophage community to those of samples from Europe. Our work is an important characterization of an economically critical insect species and shows how bacteriophage communities can contain highly conserved individuals and be highly variable in composition across a wide geographic range.

**KEYWORDS** bacteriophage assembly, bacteriophages, honey bee, metagenomics, microbial ecology, microbiome

Address correspondence to James T. Van Leuven, jvanleuven@uidaho.edu.

The authors declare no conflict of interest.

Honey bees (*Apis mellifera*) are the primary pollinators in agriculture, providing billions of dollars per year in pollination services. Like all animals, honey bees exist in a close-knit relationship with the microorganisms that live on and in them (1). These include pathogens, an essential gut bacterial community, recently described bacteriophages (phages), and microorganisms living in the hive environment. These microbes are important in determining the health of honey bees and may offer a means to

prevent disease, but much remains unknown about the ecological and evolutionary forces influencing the composition of the bee microbiome.

The honey bee gut microbiome is relatively simple and specialized compared to those of many animals. *A. mellifera* and *Apis cerana*, which diverged about 5 million years ago, share five bacterial phylotypes (>97% 16S sequence identity) that are stable in relative abundance in bees across time and geographic range. These five "core" phylotypes include bee-specific *Bifidobacterium*, *Snodgrassella*, *Gilliamella*, *Bombilactobacillus* spp. (previously referred to as *Lactobacillus* Firm-4), and *Lactobacillus* nr. *melliventris* (previously referred to as *Lactobacillus* Firm-5) (2, 3). Three additional phylotypes, *Bartonella apis*, *Frischella perrara*, and *Commensalibacter* sp., are often found in *A. mellifera* but not *A. cerana*. At the finer taxonomic level of an ~90% ANI (average nucleotide identity between whole genomes), the five core phylotypes break down into separate clusters of strains, referred to as "sequence-discrete populations," that can be specific to either *A. mellifera* or *A. cerana* (4). These clusters include closely related bacterial strains that likely represent distinct species, most of which lack formal nomenclature. Two species of *Gilliamella* (*G. apicola* and *G. apis*) have been named (5). These related species coexist in individual bee guts, and differences in their gene repertoires suggest distinct ecological niches within the bees.

The genomes of some bee gut bacteria contain an abundance of genes involved in carbohydrate metabolism and sugar transport (6, 7). These enrichments suggest that bee gut bacteria have a role in digestion and nutrient availability, compensating for processes that the animals are not able to perform on their own. *Gilliamella apicola* is responsible for pectin degradation. *Bifidobacterium*, *Lactobacillus* nr. *melliventris*, and *Bombilactobacillus* spp. are responsible for the digestion of components of the outer pollen wall and coat, such as flavonoids, $\omega$-hydroxy acids, and phenolamides, and for the digestion of hemicellulose and the utilization of the degradation products (8). *Snodgrassella* respiration creates an anoxic environment in the hindgut, allowing fermentation (9). Disturbances in these microbial communities, termed dysbiosis, can be detrimental (10–13). A number of studies have linked shifts in microbial composition to changes in a variety of host functions (14). Like many bacteria, CRISPR elements are present in the genomes of several honey bee gut symbionts, suggesting that bacteriophages are part of the honey bee gut microbiome (15).

Bacteriophages influence microbial communities in many ways (16). Phages facilitate nutrient cycling through host lysis (17), transfer genetic material between hosts (18, 19), harbor substantial metabolic gene repertoires (20, 21), interact directly with animal immune systems (22–25), and engage in antagonistic coevolution with their bacterial hosts, impacting molecules on the surface of the cell membrane (26, 27). Although the importance of phages in microbial communities is established, much remains to be learned about phage ecology, and much of the current research is focused on the human gut and aquatic ecosystems (28). Phage populations in many other important animal-associated microbial communities offer interesting study systems that will improve our understanding of the role of phage microbial ecosystems (22, 29–31).

Two recent papers describe phages in the honey bee gut (32, 33), providing insight into the potential influences of phages on bee gut bacteria. Bonilla-Rosso et al. (33) sequenced phage particles from two Swiss bee colonies, analyzed total metagenomes from Swiss and Japanese bees, and isolated a few phages on cultured host cells. Deboutte et al. (32) deeply sequenced phage particles from 102 hives from across Belgium. Both studies found a bee phage population that is diverse and largely unclassifiable (34). The majority of phages in the bee gut are predicted to be virulent (lytic) and mainly use *Gilliamella*, *Lactobacillus* nr. *melliventris*, and bifidobacteria as hosts. Most interesting is the diversity of phages observed in both studies. Whereas prophages were largely conserved across the 102 Belgian samples and in Bonilla-Rosso et al.'s metagenomic sequencing across time, the virulent phage population was variable between samples. Many clusters of closely related phages were also found in both

**TABLE 1** Metagenomic assembly metrics

| Platform | Amt of input data (Gb) | Read length | No. of contigs | $N_{50}$ (kb) | Largest contig (kb) | Mean coverage |
|---|---|---|---|---|---|---|
| Illumina (WGA)[a] | 0.7 | 150 PE | 16,448 | 3.2 | 141 | 1.5× |
| PacBio (WGA) | 62 | 4,541 ($N_{50}$) | 9,500 | 14 | 149 | 205× |
| PacBio (no WGA) | 13 | 4,593 ($N_{50}$) | 2,541 | 34 | 462 | 77× |

[a]WGA, whole genome amplification.

studies. For example, 14 bifidobacterial phages were isolated, cultured, and sequenced by Bonilla-Rosso et al. These phages clustered into only 5 groups where the ANI within each group was always more than 83%. Combined, these results point toward a highly dynamic and/or rapidly evolving phage population in the honey bee gut.

Honey bees provide an ideal model for testing how phage and bacteria interact with animal hosts because of the ease of maintaining, manipulating, and reproducing large groups in controlled environments (35). However, the temporal and geographical variations in the honey bee phage population must first be understood. We sequenced phages from a colony of U.S. honey bees and compared the phage community to those from the two other recently published studies of European honey bee colonies and found that bee phages are mostly novel, are more similar to one another than to phages from other environments, are predicted to infect only a subset of the honey bee gut bacteria, and have sequence diversity that reflects host diversity. Surprisingly, a few phages were highly conserved in bees from all three countries. This combination of a small set of conserved phages with a larger, highly individualized set of phages highlights a need for temporal characterization of animal-associated phages and *in vitro* testing of phage host range to understand animal microbiomes.

## RESULTS

**Identifying the best viral metagenome assembly.** Sequencing methods influence viral genome assembly and community characterization (36, 37). As a proof of concept, we sequenced whole-genome-amplified viral DNA extracted from Texas honey bees using Illumina MiSeq short-read technology. The same sample was then sequenced using PacBio Sequel II long-read technology. Because whole-genome amplification (WGA) is known to introduce biases into viral community descriptions (38, 39), we also sequenced a nonamplified sample using PacBio. The assembly of only Illumina MiSeq reads (WGA sample) resulted in 16,448 contigs with an $N_{50}$ of 3,209. The WGA sample sequenced using PacBio assembled into 9,500 contigs with an $N_{50}$ of 14,739. The non-WGA sample was assembled into 2,541 contigs with an $N_{50}$ of 34,304 bp (Table 1). These three assemblies contained overlapping sets of contigs, although the Illumina assembly was much more fragmented than either PacBio assembly, likely due to the shorter read lengths and the relatively small amount of sequencing done on this library.

Given that the non-WGA PacBio assemblies had sufficient read coverage and should be less prone to artifacts introduced by the WGA (40), we analyzed only the non-WGA assembly. Of the 2,541 contigs, 412 were putatively circular molecules according to the assembler (Flye). These circular contigs ranged from 502 bp to 107,828 bp in length and 3× to 943× in coverage (Fig. 1). In addition to the 2,541 contigs, 277 plasmids were assembled by Flye. These plasmids ranged from 900 bp to 33,327 bp in length and from 1× to 483× in coverage. Several of these "plasmids" were identified as putative phages in downstream analysis and were subsequently included in all analyses.

**Identification of bacteriophage in Texas honey bees.** Of the 2,541 contigs in a non-WGA PacBio assembly, we identified 477 putative phage sequences (genomes) that were longer than 1,000 bp and were identified by at least three phage-finding programs. We found that PPR-Meta identified the most phages, followed by VIBRANT, DeepVirFinder, VirSorter2, PHASTER, PPR-Meta, and VirFinder (see Fig. S1 and Table S1 in the supplemental material). While PPR-Meta and VIBRANT identified more phages than the other programs, the phages identified by the other programs were identified

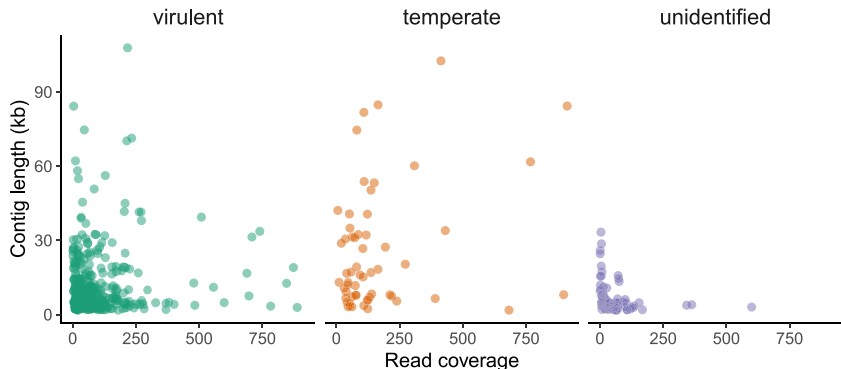

**FIG 1** Length and abundance of contigs identified as bacteriophage. Lifestyle was determined using VIBRANT.

by most algorithms, leading us to believe that PPR-Meta and VIBRANT are the least stringent (or most sensitive) classifiers. To test PPR-Meta, we compared the output from our virome assembly to computationally generated contigs of the same length made from the *A. mellifera* genome. PRR-Meta identified phages in 42% of the metagenomic contigs versus only 5% in the size-matched bee genome contigs. Of the 477 putative phages, 58 were temperate, 325 were virulent, and 94 were unclassified by VIBRANT. The lengths and coverages of temperate and virulent phages were quite variable (Fig. 1), as were the assembly qualities reported by PHASTER, VIBRANT, and CheckV. PHASTER classified 23 (18%) intact, 88 (68%) incomplete, and 18 (14%) questionable genomes. VIBRANT reported 24 (6%) high-quality, 32 (8%) medium-quality, and 327 (85%) low-quality phage genomes. CheckV identified 3 (1%) complete, 53 (11%) high-quality, 43 (9%) medium-quality, 301 (66%) low-quality, and 77 (16%) undetermined viral contigs (Fig. S2). These numbers seemed low considering that 214 (45%) of the contigs were identified as complete circular contigs by Flye. Combining quality and identification metrics from PHASTER, VIBRANT, and Flye resulted in a list of roughly 150 phage genomes in our data set that were of high quality (circular/complete) and had identifiable hosts or matches to phages in public repositories. The remaining 300 or so contigs (477 minus 150) had either lower-quality genome completeness metrics or no host/taxonomic designation. These contigs were still identified by at least three phage-finding algorithms, so we included them in our analyses. We also used CheckV ANI clustering to identify 50 potentially redundant contigs. These contigs are identified in Table S2 and are available in GenBank and GitHub.

As was previously found by Bonilla-Rosso et al. and Deboutte et al., most phages in *A. mellifera* could not be taxonomically identified. We classified only 66 of the 477 (13%) viral contigs to the family level. These classifications were based on vCONTACT2 clustering (Fig. 2) to the viral RefSeq database and phage sequences from Bonilla-Rosso et al. and Deboutte et al. (32, 33). Bonilla-Rosso et al. and Deboutte et al. classified 24% (28/118) and 26% (73/273) of their viral clusters, respectively. The distributions of contigs into the virus families were comparable among the three studies, with the majority of phages belonging to *Siphoviridae* and *Myoviridae* and with rare observations of *Podoviridae* and *Microviridae* (Fig. 3). The read coverage of individual contigs was highest for *Siphoviridae* and *Myoviridae*, further confirming their dominance in the bee microbiome. The predicted hosts of some of these phages were *Lactobacillus* (including the newly designated genera *Bombilactobacillus* and *Apilactobacillus*). Unlike the studies by Bonilla-Rosso et al. and Deboutte et al., we did not find any *Inoviridae*, but we did find four contigs classified as *Gokushovirinae*. We presume that these single-stranded DNA (ssDNA) phages were detected in an intermediate double-stranded DNA (dsDNA) form, and thus, their abundance in the sequencing data from us and Bonilla-Rosso et al. (33) is likely not an accurate measure of the abundance of mature

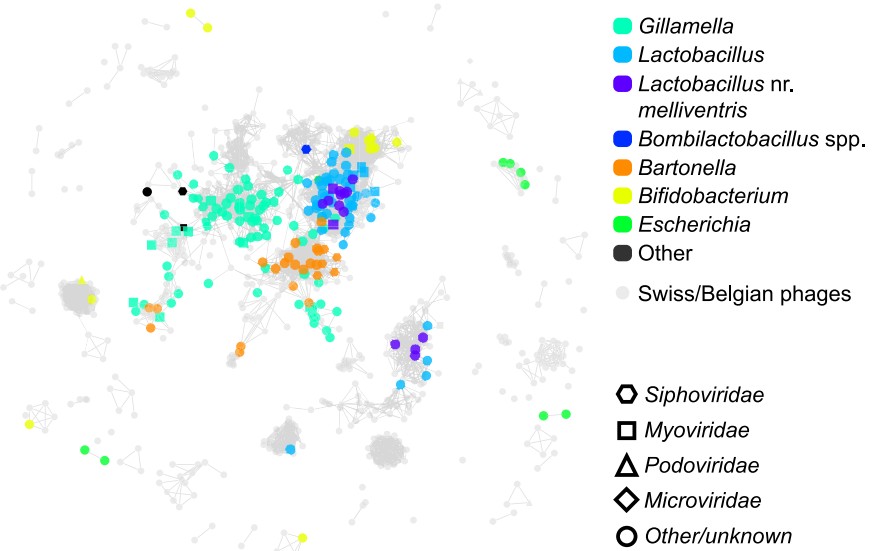

**FIG 2** Network diagram showing clusters of phages from honey bees, classified by host (top) and viral (bottom) taxonomies. Clustering is based on the similarity between protein-coding genes among viral contigs. Phages from the studies of Deboutte et al. (32) and Bonilla-Rosso et al. (33) are shown in gray.

virions. The *Gokushovirinae* contigs all belong to one cluster, two of which share 99.92% identity across their ~5-kb genomes. Although this extreme similarity between phages was uncommon in our data, many clusters contained groups of phages sharing high sequence similarity. The vCONTACT2 network (Fig. 2) roughly illustrates the size and connectedness of these clusters.

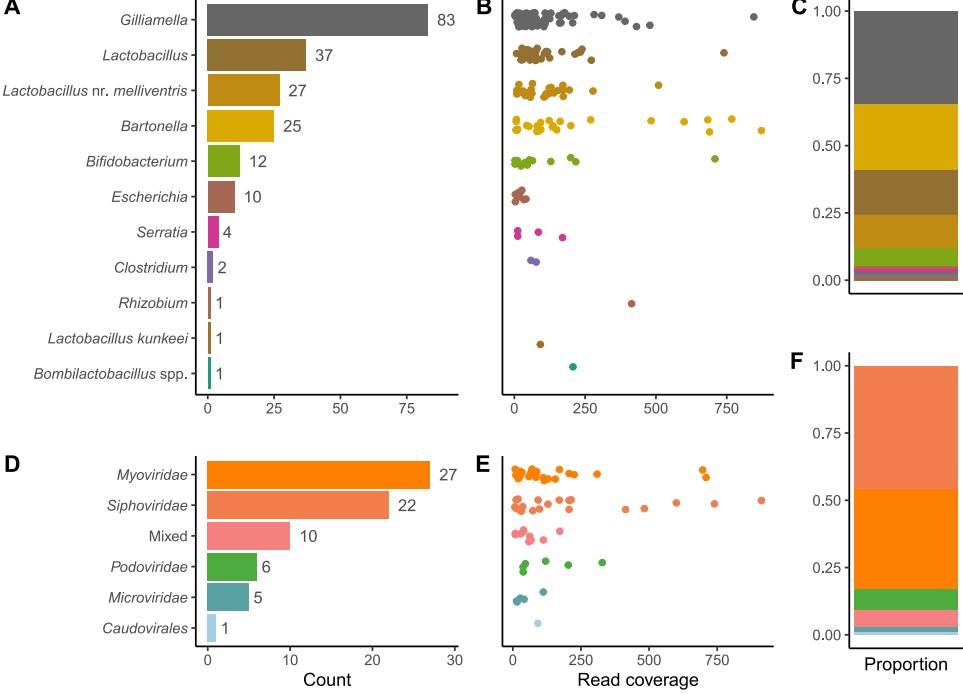

**FIG 3** Characterization of the bee phage community. (A and D) Numbers of phages identified. (B and E) Read coverage of individual phage contigs. (C and F) Relative abundance (read coverage) of phages by host usage and taxonomy.

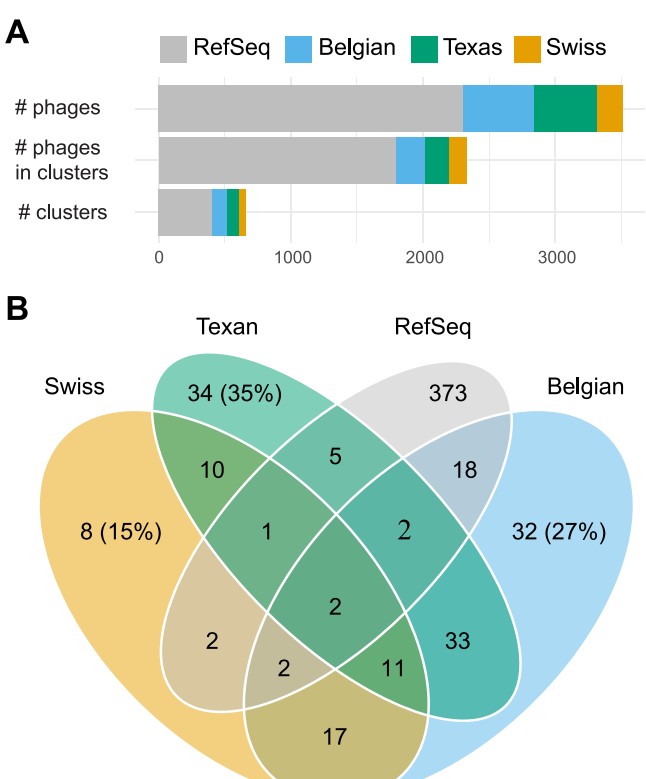

**FIG 4** Phages from bees more readily cluster with one another than with phage sequences in reference databases. (A) Incorporation of individual phage contigs into clusters. (B) Composition of the final set of clusters, based on which phages are in these clusters. For example, there are 8 clusters containing phages from only the Swiss data set. There are 13 (11 + 2) clusters with phages from all three honey bee phage viromes.

We were able to predict the hosts of 203 of the 477 phages (Fig. 3) in our sample using a hierarchical scheme of searches. First, we performed a BLASTn search against CRISPR spacer sequences from honey bee gut bacteria reported by Bonilla-Rosso et al. Second, we performed a BLASTn search against a recently compiled CRISPR sequence database (41). In both CRISPR searches, we required 100% matches between our contigs and spacer sequences. Third, we clustered viral clusters (VCs) from our study and those of Bonilla-Rosso et al. and Deboutte et al. and then used host identifications from these studies. Finally, we ran our contigs through the vHULK prediction tool (42). In only a few cases did these methods disagree, but when they did, we chose the host according to the order outlined above (Table S2).

A total of 27 contigs were predicted to infect *Lactobacillus* nr. *melliventris*, and only 1 was predicted to infect *Bombilactobacillus*. An additional 37 contigs were classified to *Lactobacillus* broadly instead of one of the two dominant clades of *Lactobacillus*-related bacteria within honey bees (*Bombilactobacillus* and *Lactobacillus* nr. *melliventris*). Phages predicted to infect *Gilliamella* were most numerous, with 83 of the 203 phages with predicted hosts belonging to *Gilliamella*. In contrast to the results of Bonilla-Rosso et al. and Deboutte et al., we found a number of microviruses that are predicted to infect *Escherichia coli*. We inspected these viral contigs with interest as we were worried about potential contamination from ΦX174 strains regularly used in the same laboratory. However, we found that there were indeed a number of microviruses (subfamily *Gokushovirinae*) that were related to *Gokushovirinae* found in other bees (43) as well as one with ~75% BLASTn matches to phages from other environments. None shared sequence identity with any of the microvirus strains cultured in our laboratory.

**Conservation in the global honey bee phage population.** Using vCONTACT2, we clustered the 477 phages from Texas honey bees with the VCs described by Deboutte

et al. and Bonilla-Rosso et al. (Fig. 2) and publicly available phage sequences (Fig. 4). Most of the 1,203 phages from the three honey bee phage metagenomic assemblies were identified as singletons or outliers by vCONTACT2 (Fig. 4; Fig. S3). These novel phages numerically dominate the bee phage community (Fig. 4). The 1,203 phages from honey bees clustered into 175 unambiguous clusters, 98 of which contained phages from Texas bees. A total of 184 of the 477 Texas bee phages clustered with other phages (Fig. 5). Bee phages from the three honey bee viromes clustered with one another more often than with phages from the viral RefSeq database (Fig. 4). Thirteen clusters contained phages from all three honey bee phage viromes. Ten of these are predicted to be virulent phages, and three are predicted to be temperate. These phages included podoviruses and myoviruses of bifidobacteria, siphoviruses and caudoviruses of *Lactobacillus* nr. *melliventris*, and myoviruses, caudoviruses, and sipho-viruses of *Gilliamella*. Thirty-five phage clusters were uniquely shared between the Belgian and Texas samples, 17 were shared between the Belgian and Swiss samples, and 11 were shared between the Swiss and Texas samples. Thirty-nine clusters from Texas honey bees did not cluster with the European bee phages, but 5 of these clustered with phages in RefSeq. Many clusters that included phages from RefSeq included more than one RefSeq phage, resulting in large cluster sizes (Fig. 5). The largest of these clusters contained four microvirids that we identified in Texas bees but that were absent from the European samples. This cluster (VC_196) contained four *Gokushovirinae* genomes, all ranging in size from 4.5 to 5.5 kb. Two of these differ from one another by only a few hundred nucleotides, but the others were more distantly related. All harbor genes to make the major and minor capsids, internal scaffolding, replication proteins, and ssDNA synthesis proteins. While vHULK suggested an *E. coli* host, vCONTACT2 clustered these contigs with phages identified previously in honey bees (44). Clusters containing closely related phages were common in all three data sets. The largest clusters of phages with representatives from all three bee phage viromes are mostly myoviruses that infect *Gilliamella*, *Lactobacillus* nr. *melliventris*, and *Bifidobacterium* (Fig. 2, 5, and 6). The sequence length and synteny of phages in these clusters were quite variable. A representative set of myovirus genomes illustrates the diversity observed in the vCONTACT2 clusters (Fig. 6). In the 13 clusters, the most similar phages have an average of 93% amino acid identity (AAI) across their protein-coding genes (Fig. S5 and Table S5). The most dissimilar phages have 32% AAI (45% average similarity) but were mostly confined to one cluster (VC_78) of *Lactobacillus* nr. *melliventris* phages. Of the 13 clusters that contained representative genomes from all three data sets, only 5 contained Texas phages designated circular or "high quality."

**Gene content of bacteriophage in Texas honey bees.** A total of 15,228 protein-coding genes (coding DNA sequences [CDSs]) were identified in the 477 phage contigs using multiPhATE2. Of these, 5,983 were >100 amino acids in length, and 3,807 had some functional annotation (i.e., not "hypothetical protein" or "phage protein"). Forty-eight phage contigs contained CDSs with only hypothetical protein annotations. Of the 456 genes in these 48 phage contigs, 192 were >100 amino acids in length. The number of CDSs in these contigs ranged from 3 to 31. At least some of these contigs are likely complete phage genomes containing only genes of unknown function.

The vast majority of genes with sequence matches to protein databases were phage structural proteins (capsid, tail, baseplate, portal, and scaffolding). Also among the most common types of proteins were terminase, tape measure, integrase, repressor, polymer-ase, helicase, nuclease, DNA methyltransferase, and endolysin proteins (Table S3). About half of the viral contigs contain at least capsid and tail/spike proteins. Other genes such as those encoding RecA, virulence factors, superinfection exclusion, and "plasmid pro-teins" were common among the phage contigs. Metabolic genes were rare, although a few genes seemingly involved in queuosine, teichoic acid, and riboflavin biosynthesis were present in four viral contigs. Queuosine biosynthesis genes have been found on phage genomes and may be involved in protection from genome degradation by the host (45). In addition, the annotations of 372 of the 3,807 CDSs were difficult to parse

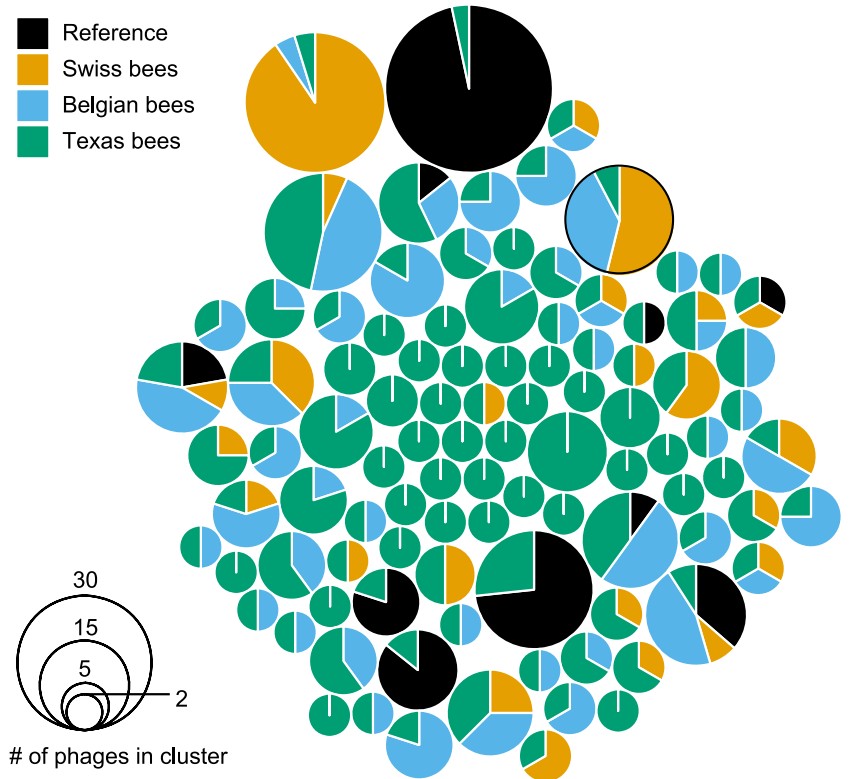

**FIG 5** Phages from Texas honey bees mostly form small clusters of only Texas honey bee phages. A total of 98 vCONTACT2 clusters containing 184 Texas honey bee phage contigs are shown. The circle size is proportional to the number of phage genomes in a cluster. The proportion of phage genomes from each of the four sources of phage genomes is shown. VC_106 (Fig. 6) is circled.

(complex description) and/or not shared among any other CDSs (Table S3). The length of these proteins was not shorter than the length of proteins with easily discernible functions, suggesting that additional functional capacity is hidden in these genes.

**Bacterial contamination of the bee virome.** Contaminating bee gut bacteria were detected in the viral sequencing data at low levels. A total of 1,875 of 3,379,211 (~0.1%) of the reads were mapped to full-length bacterial small-subunit (SSU) rRNA sequences from the SILVA database using SortMeRNA. Grouping these bacteria by genus resulted in proportional abundances of 36% *Lactobacillus* nr. *melliventris*, 17% *Bombilactobacillus*, 15% *Gilliamella*, 13% *Bartonella*, 13% *Bifidobacterium*, 2% *Snodgrassella*, 1% *Mesorhizobium* (presumably from pollen), and ~0.5% *Commensalibacter* (Table S4). Given that phages regularly encapsulate bacterial DNA, we cannot say if these contaminating sequences are in phage capsids or simply were not digested during DNase treatment. Read coverages for bee-associated bacterial SSU rRNAs ranged from about $50\times$ to $700\times$, completely within the range of phage genome read coverages. Bonilla-Rosso et al. found virulent phages to be much more abundant than temperate phages (90% versus 10% of reads) in virus-enriched samples, whereas in total microbial metagenomes, the pattern was opposite, suggesting that viral enrichment protocols reduce potential bacterial contamination and improve the detection of phage-encapsulated DNA. To estimate how well our enrichment protocol performed, we used ViromeQC to calculate a virome enrichment score (46). The enrichment scores for Illumina, PacBio, and no-amplification PacBio reads were 24.4, 4.8, and 1.8, respectively, indicating at least some phage enrichment. The large discrepancy among these scores is possibly due to having no PacBio reads aligned to either SSU rRNA or large-subunit (LSU) rRNA genes. The rates of alignment to bacterial markers for Illumina, PacBio, and no-amplification PacBio reads were $1.7E-2$, $9.9E-2$, and $2.7E-1$, respectively. Illumina reads had SSU rRNA and LSU rRNA alignment rates of $1.9E-3$ and

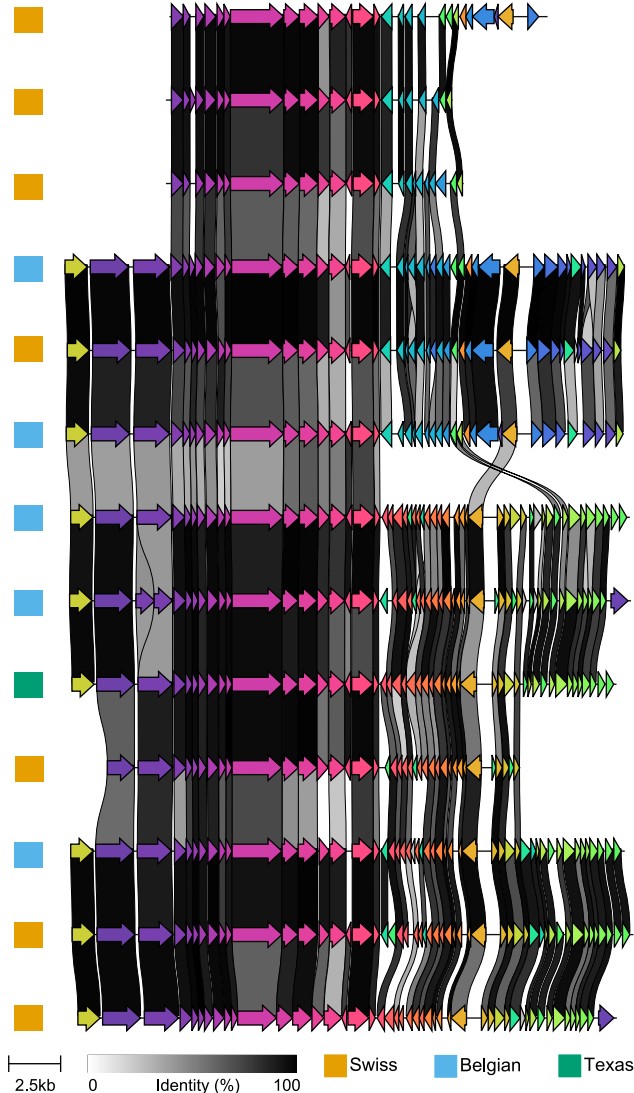

2.5kb   0   Identity (%)   100   ■ Swiss   ■ Belgian   ■ Texas

**FIG 6** Alignment of bifidobacteria infecting myoviruses from cluster VC_106. Clinker (72) was used for genome ordering and visualization. A cutoff of 30% sequence identity was used for plotting connections between genes.

5.4E−3, respectively. It is unclear if PacBio reads are suitable for use with ViromeQC. Our libraries were all made from the same virome-enriched sample.

## DISCUSSION

Much remains to be learned about the ecology and evolution of bacteriophages in the microbial communities of animal digestive tracts. Even in well-studied microbial communities, like in the human gut, key characteristics of the microbial ecological networks are not well understood. What phages infect what hosts? How many hosts are killed by phages? How often do temperate phages become lytic? Do phages drive bacterial diversification? Honey bees offer a promising system to study these questions. The bacterial composition of the honey bee gut is one of the most well characterized of all animals (1, 6, 14, 35). The honey bee microbiome differs between castes (47, 48), across seasons (49), and between regions (50) and changes in response to environmental conditions (10, 11). Moreover, these bacteria have beneficial impacts on their bee hosts (8) and are culturable, providing the opportunity to develop microbiome-mediated treatments for disease (51).

Our comparison of the three available honey bee gut viromes provides a first look at widespread geographic variation in the honey bee phageome, giving insight into

the role of these viruses in animal microbiomes. We found that 13 phage clusters are shared among bees from Switzerland, Belgium, and the United States, suggesting that a small set of phages is widely distributed in domesticated honey bees. This set of phages is complemented by a large variable population. Interestingly, the phage community in the human digestive tract is also highly distinct among individuals, with the exception of crAssPhage and perhaps a few other phages that are frequently observed in many individuals (52). crAssPhage is ubiquitous in humans and several groups of primates (53). While the crAssPhage group is diverse, some strains are highly conserved and have remained colinear for millions of years, highlighting a paradox in phage biology: while most phages evolve quickly and are incredibly diverse, some species can be highly conserved and widely distributed.

Bacteriophages affect bacterial communities in a number of ways. Virulent phages lyse their host bacteria, reducing their hosts' abundance and causing the release of cellular nutrients into the environment. Temperate phages integrate into their hosts' genomes, often providing temporary benefits to their lysogens. Both virulent and temperate phages shuffle genes via horizontal gene transfer, harbor auxiliary metabolic genes, and affect the evolution of their hosts. In all three honey bee viromes, virulent phages dominate the phage community in abundance and diversity. Virulent phages are often abundant in dense, productive microbial communities such as in animal (including human) digestive tracts (22, 54), moist soils (55), and animal waste slurry (36), etc. (30). In these types of microbial communities, phages likely play a large role in the ecology of the resident bacteria. However, we also note that strict classification of phages as virulent or temperate can be challenging given the presence of intermediate lifestyles such as pseudolysogeny, the occasional integration of phages traditionally identified as virulent, and the tendency for classifiers to identify phages as virulent (22, 54, 56). Still, the bee gut microbial community seems to be one that supports a large population of virulent phages that likely play an important role in the microbial ecology of the bee gut.

In our study, the most abundant (measured by relative read coverage) phages are predicted to infect not the most abundant bee gut bacteria but rather *Bartonella* and *Gilliamella*. Compared to our study, Bonilla-Rosso et al. and Deboutte et al. found a high proportion of phages infecting *Bifidobacteria* and *Lactobacillus*. None of the three studies (Texas, Belgium, and Switzerland) quantified the relative abundance of bacterial hosts, which could account for the differences in phage abundance. Rather, host abundance is inferred from many studies on bee gut bacterial composition. The seasonal fluctuations in the absolute abundance of bacteria in bee guts can be substantial (10- to 100-fold), although the compositional frequencies do not change so drastically (49). The Texas and Swiss hives were sampled in January, while the Belgian hives were sampled in autumn. Bacterial populations change from being primarily *Bartonella* and *Lactobacillus* nr. *melliventris* in the winter to being *Gilliamella* and *Snodgrassella* and/or *Frischella* during foraging seasons (49). No phages predicted to infect *Commensalibacter* sp. were found in any of the three studies. Phages predicted to infect *Snodgrassella* or *Frischella* were found by Bonilla-Rosso et al. but were rare and were not found by Deboutte et al. or in our study. In humans, phage and host abundances can be well correlated (54, 57). Similar correlations were observed in wastewater treatment plants (58) and seawater (59). In future studies on the bee microbiome, it will be interesting to measure bacterial and phage abundances over time to test for correlated dynamics between phage and host. Given the current data, it seems that some prevalent bacteria in the bee gut are entirely free from predation by phages.

The hosts for about 25 to 50% of bee phages, including one of the 13 phages present in all three bee phage studies, could not be identified, highlighting a common result in phage metagenomic studies: phage classification can be difficult. Even in the well-studied human gut, much about phage ecology remains to be learned. Currently, crAssPhages (first described in 2014) are the most abundant and diverse group of phages (60, 61). Different crAssPhage genera exist across the globe and

probably utilize hosts belonging to the phylum *Bacteroidetes* (53). However, finer-grain taxonomic determination of the host has not yet been resolved except in a few instances (60). Either *Microviridae* or members of a crAssPhage-like group, *Gubaphage*, are likely the second most abundant phages (22, 25, 54), although other recently described new families are also abundant (31). Hosts for *Microviridae* and *Gubaphage* remain elusive but likely include bacteria from broad taxonomic groups. Most likely, phage abundance is dependent on many factors, such as the number of available hosts, the strength of defenses employed by the host, the presence of competing phages, and the number of alternative hosts. Rapidly improving sequencing technologies (e.g., Hi-C and single cell) and host determination algorithms will help facilitate improved predictions of phage hosts and, thus, a better understanding of phage ecology.

Phages infecting *Gilliamella*, *Lactobacillus* nr. *melliventris*, *Bartonella*, and *Bifidobacterium* make up roughly 40%, 32%, 11%, and 6% of the unique types of assigned phages identified in our study, respectively. Many of these phages share enough sequence identity to form groups of phage clusters that are similar to one another and discrete from other bee phages. Bee gut bacteria are similarly diverse. *Gilliamella*, *Lactobacillus* nr. *melliventris*, *Bombilactobacillus*, and *Bifidobacterium* are present in bee populations as multiple discrete clusters of related strains or species that are diverged by at least 10% across the genome (nucleotide sequence identity) (62, 63). While *Bartonella* populations also have high diversity, they do not segregate clearly into discrete clusters, based on current sampling. Individual honey bees are colonized by a small subset (sometimes just one) of the many closely related strains present in a community. Since we sequenced a pool of 75 bees, we may have observed a higher diversity of phages than what is present in individual bees. Bonilla-Rosso et al. also sequenced pools of ~100 hindguts, while Deboutte et al. used smaller samples of 6 bees (2 bees from 3 hives each), although this required whole-genome amplification. Deboutte et al. found that few phages were shared among the 102 samples that they collected. The maximum number of phages shared between any two samples was 15. However, 20 phages were shared among at least 5 samples. Their sampling locations spanned the entire northern region of Belgium and were collected over 2 years. The most similar phages shared among all three bee viromes have about 93% average amino acid identity (AAI) across the genome (see Table S5 in the supplemental material). Most of the 13 clusters have phages with AAIs above 90%. The most dissimilar phages (~32% AAI and 45% average similarity) were phages of *Lactobacillus* nr. *melliventris*, a diverse group of hosts. It was recently shown that closely related strains of *Lactobacillus* nr. *melliventris* coexist in the honey bee gut through niche partitioning of pollen metabolism (64). Whether or not phages specialize on these functionally divergent bacteria and if phage host range evolution affects these bacterial communities remain interesting questions. Ecological models (65) and empirical studies (66) of phage and hosts show that the stable coexistence of phages with overlapping host ranges can occur under certain conditions, specifically when there is a fitness trade-off between generalist and specialist strategies or in spatially structured gut environments (67). As we learn more about phage ecology in animal guts, the bee gut microbiome provides a unique and convenient system to experimentally explore microbial ecosystems.

## MATERIALS AND METHODS

**Preparation of sequencing libraries.** Honey bees were sampled from the rooftop hives from UT Austin in January 2020. The digestive tracts of 75 *Apis mellifera* honey bees were dissected in cold phosphate-buffered saline (PBS) after euthanizing them at −20°C for 20 min. As previously described, the digestive tract is easily removed by pulling on the stinger. The remaining tissues were preserved in 95% ethanol (EtOH). Approximately 7 mL of PBS and dissected guts were homogenized using an ice-cold mortar and pestle. The homogenized material was centrifuged at 5,000 × *g* for 5 min to remove cellular debris. The supernatant was pushed through a 0.22-$\mu$m filter. The filters clogged after about 2 mL, so 3 filters were used. The filtrate was treated with 0.25 U/$\mu$L DNase and RNase for 6 h at 37°C to remove nucleic acids not protected by viral capsids. The nucleases were deactivated for 10 min using 0.5 M EDTA. Nucleic acids were extracted using phenol-chloroform extraction, followed by two additional chloroform extraction steps. The volume of the aqueous layer was kept at ~5 mL by adding 10 mM Tris buffer. Ethanol precipitation using 2.5 volumes of EtOH and a 1/10 volume of

sodium acetate was used to purify nucleic acids. Contaminants were further removed using 1.5 volumes of magnetic beads (MagBio HighPrep). The final sample was eluted in 50 $\mu$L elution buffer (10 mM Tris-Cl). The fragment analyzer (Agilent) trace showed dilute (0.1 ng/$\mu$L) DNA with a size range of 100 to 150 kb. Whole-genome amplification (GE illustra GenomiPhi V2) was performed on 1 $\mu$L purified DNA. Amplified DNA was purified with 1.5 volumes of magnetic beads (MagBio HighPrep) and sent to the MiGS Genome Sequencing Center (Pittsburgh, PA) for Illumina sequencing (150-bp paired end [PE]). Both the unamplified and WGA samples were sequenced by PacBio Sequel II at the Genomics Resources Core at the University of Idaho.

**Sequencing.** An Illumina sequencing library was generated by MiGS for the whole-genome ampli-fied (WGA) DNA according to Illumina Nextera kit protocols and sequenced on the NextSeq 550 platform to generate 2,524,978, 150-bp PE reads.

Two PacBio libraries were made at the Genomics and Bioinformatics Research Core at the University of Idaho according to standard PacBio library preparation protocols. Totals of 3,881,928 and 11,631,485 continuous long reads (CLR) reads were generated after demultiplexing using SMRT Link (Lima) for non-amplified and WGA samples, respectively. The third library contained an *E. coli* strain closely related to ATCC 13706. We removed contigs matching this genome using bwa-mem.

**Genome assembly and analysis.** Illumina reads were quality filtered and adaptor trimmed using fastp v0.20.0 (–detect_adapter_for_pe) and assembled using SPADES v3.9.0 (–careful). PacBio CLR reads were assembled using Flye v2.7.1 with the –meta options.

Flye-assembled contigs that were at least 2,000 bp long (minimum input contig length) were ana-lyzed in PHASTER through the Web server to identify possible phage sequences. Positive hits identified as incomplete, questionable, or intact were compared to other phage detection software results. What the Phage was run on contigs of >1,000 bp in length (nextflow run replikation/What_the_Phage -r v0.9.0 –cores 8 –fasta polished_1.fasta -profile local,docker). The resulting output files were combined with PHASTER results using R.

Nearly identical contigs were collapsed using cd-hit (cd-hit -i contigs_plasmids_phage.fa -c .99 -T 4 -o contigs_plasmids_phage_cdhit99.fa). We also tested how further leniency in collapsing parameters would reduce the number of contigs and found that using a cd-hit cutoff of 95% identity reduced the 477 phage contigs down to 430 viral contigs. We chose to analyze phages at the 99% cutoff. In addition, we performed ANI clustering using the aniclust.py script in CheckV. At the 99% level, this reduced the 477 contigs to 413 (see Table S2 in the supplemental material).

Phage hosts were identified using a number of tools. First, BLASTn was used to search for Texas phage contigs in CRISPR spacer sequences from bee gut bacteria reported previously (see supplemental Data Set S4 in reference 33) (blastn -ungapped -dust no -soft_masking false -perc_identity 100 -outfmt 6 -num_threads 4). Second, Texas phages were compared to the DASH CRISPR database using BLAST (blastn -db SpacersDB.fasta -query contigs_plasmids_phage.fa -ungapped -dust no -soft_masking false -perc_identity 100 -outfmt 6 -num_threads 4). Third, vCONTACT2 was used to cluster phages and iden-tify putative hosts. Finally, vHULK v0.1 was run on contigs of >5 kb in length (default flags, "final predic-tion" used for host).

A multifasta file was generated containing 477 Texas bee phages, 190 phages from the study by Bonilla-Rosso et al. (33), and 537 phage contigs from the study by Deboutte et al. (32). They were clustered using vCONTACT2 in the CyVerse Discovery Environment (MCL clusterOne Diamond RefSeq V85 E value = 1E−4). The vCONTACT2 network file was visualized in Cytoscape 3.8.2 with characteris-tics from the combined phage identification protocol (PHASTER and What the Phage). The results were visualized with and without (Fig. 2) 85 RefSeq viral sequences. The clusters were arranged using the Prefuse force-directed OpenCL layout. Viral taxonomy was determined based on the known taxon-omy of other viruses in the vCONTACT2 clusters. Contigs with the taxonomic designation "mixed" clustered with phages from different families, and the correct group could not be easily determined using BLAST searches of the nonredundant (nr) database. multiPhATE2 was used to annotate 477 putative phage genomes with the following parameters: phanotate_calls='true', prodigal_calls='true', glimmer_calls='true', primary_calls='phanotate', blastp_identity='50', blastp_hit_count='5', blastp='true', phmmer='true', pvogs_-blast='true', and phantome_blast='true'. The gene descriptions from the BLASTp and hmm hits were searched in R. Gene categories were roughly taken from the list of common phage genes reported previ-ously (68).

Bacterial contamination was detected by extracting PacBio reads with nucleotide similarity to small-subunit RNA sequences from the SILVA database. To reduce computational requirements, only reads that were >3 kb in length (~90% of the reads) were used (3,379,211 total reads). A condensed database was generated, clustering SSU rRNA sequences at 96% similarity, according to the phyloFlash protocol (69). Reads containing SSU rRNA were pulled from metagenomic sequencing using SortMeRNA (70), using default parameters. The distribution of BLAST hit lengths and identities is shown in Fig. S4. Only full-length (1,400 to 1,700 bp) matches were analyzed for estimating bacte-rial contamination in the virome. A total of 200,000 reads from each PacBio data set and 2,000,000 forward reads from our Illumina data set were run through ViromeQC (-w environmental) to estimate virome enrichment scores. The PacBio data were subsetted because the full data set used all our server's RAM (192 Gb). However, no PacBio reads were aligned to rRNA genes, even though ~700 would have been expected based on the 7.3E−3 alignment rate of our Illumina reads. MetaQUAST was used to compare assemblies from the three sequencing methods (Illumina WGA, PacBio WGA, and PacBio).

Gene alignments for phages in the 13 shared clusters were generated by reannotating phage con-tigs using PROKKA (71) to make *.gbk files, which were then used by Clinker (72) for alignment. A shell

wrapper script is included in the GitHub project. Analysis of the Clinker output was performed in R and is included in the GitHub project associated with this paper.

**Data availability.** Raw sequencing reads were deposited in GenBank under accession numbers SRR17894214 to SRR17894216 and under BioProject accession number PRJNA803764. The R code for all analyses is available in the GitHub repository at https://github.com/jtvanleuven/bee_phage.

## SUPPLEMENTAL MATERIAL

Supplemental material is available online only.

**FIG S1**, PDF file, 0.2 MB.
**FIG S2**, PDF file, 0.01 MB.
**FIG S3**, PDF file, 0.01 MB.
**FIG S4**, PDF file, 0.02 MB.
**FIG S5**, PDF file, 0.1 MB.
**TABLE S1**, XLS file, 0.2 MB.
**TABLE S2**, XLS file, 1.4 MB.
**TABLE S3**, XLS file, 1.1 MB.
**TABLE S4**, XLS file, 0.04 MB.
**TABLE S5**, XLS file, 0.1 MB.

## ACKNOWLEDGMENTS

We thank Holly Wichman, LuAnn Scott, Joanne Emerson, and Dan New for laboratory assistance and helpful discussions. We also thank the National Summer Undergraduate Research Project for matching T.J.B. with mentor J.T.V.L. and the California Alliance for Minority Participation (LSAMP/CAMP) at the University of California, Davis, for providing professional support and presentation opportunities. We thank two anonymous reviewers of the manuscript for valuable feedback and two researchers who provided constructive comments on the preprint.

Research reported in this publication was supported by the National Institute Of General Medical Sciences of the National Institutes of Health under Award Number P20GM104420, National Science Foundation (NSF) Idaho EPSCoR Program under award number OIA 1757324, startup funds awarded to C.R.M., and sequencing vouchers from the Office of Research and Economic Development at the University of Idaho. The content is solely the responsibility of the authors and does not necessarily represent the official views of the National Institutes of Health.

We declare no competing interests.

J.T.V.L. conceived and planned the experiments. T.J.B. and J.T.V.L. performed the experiments, analyzed the data, and drafted the manuscript. C.R.M. and N.A.M. provided critical resources and interpreted the results. All authors edited the manuscript.

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
