## [Reviewer comments · mSystems]

Global composition of the bacteriophage community in honey bees

Taylor Busby, Craig Miller, Nancy Moran, and James Van Leuven

Corresponding Author(s): James Van Leuven, University of Idaho

Review Timeline:

Submission Date:	September 30, 2021
Editorial Decision:	November 17, 2021
Revision Received:	March 1, 2022
Accepted:	March 2, 2022

Editor: Michela Gambino

Reviewer(s): The reviewers have opted to remain anonymous.

Transaction Report:

DOI: <https://doi.org/10.1128/mSystems.01195-21>

November 17, 2021

Dr. James T Van Leuven
University of Idaho
Department of Biological Sciences
Moscow, ID 83843

Re: mSystems01195-21 (Global composition of the bacteriophage community in honeybees)

Dear Dr. James T Van Leuven:

Thank you for submitting your manuscript to mSystems. We have completed our review and I am pleased to inform you that, in principle, we expect to accept it for publication in mSystems. However, acceptance will not be final until you have adequately addressed the reviewer comments below. Please, also make sure to add the Genbank accession numbers.

Preparing Revision Guidelines

Sincerely,

Michela Gambino

Editor, mSystems

Journals Department
Reviewer comments:

Reviewer #1 (Comments for the Author):

The manuscript by Busby and colleagues investigated the bacteriophage community associated to honeybees sampled in Texas, United States, and compared these results with the ones previously obtained on two European bee populations by Bonilla-Rosso et al., 2020 and Deboutte et al., 2020. Considering that little information is available on the structure, dynamics, and function of honeybee associated viral communities, this work greatly contributes to fill in this gap. The research question is very interesting and the work done is of high quality.

The manuscript is divided in well-balanced sections. The section "introduction" describes the state-of-the-art of the study, providing to the Readers the information needed to contextualize and understand the work. The aim of the work is clearly stated at the end of the Introduction section. "Results" and "discussion" are well presented and organized. "Materials and methods" are clear and informative to allow reproduction of the experiments.

I have only few minor comments.

L. 34: please, check the presence of an extra "and".

L. 222 please check "phagess".

L. 372 and 388: please, check the upper-case letter of "melliventris".

L. 424: Genbank accession numbers are missing.

L. 441 and 445: is "MagBio beads" the name of the kit that Authors used?

L. 453: please, check "U of I".

Reviewer #2 (Comments for the Author):

Busby et al. present the analysis of a new virome dataset from honeybee gut samples. Specifically, the authors first tested a few different approaches for sequencing the sample (including short- and long-reads, as well as with and without WGA), and then focus on the analysis of one contig collection (from non-amplified long-reads data) that is compared to two recently published honeybee virome catalogs.

Overall, I found the manuscript to be clearly written, compelling, and a clear contribution to our collective understanding of the honeybee "phageome". I only have 2 major comment that I think need to be addressed:

- The authors include a section about bacterial contamination, however I found this section to be a bit short and lacking context for a non-expert reader to interpret. I would encourage the authors to look in the literature (e.g. 10.1038/s41587-019-0334-5) to have some reference point, and discuss how the viral enrichment protocol performed in their case.

- The authors use multiple approaches for computational host prediction, which is a good idea. However, they mostly provide the consensus host prediction and not the predictions obtained with individual tools. To be able to interpret and possibly re-analyze these data, ideally the authors would provide in a supplementary table a detailed list of host prediction, with one row per "individual" prediction (e.g. vHULK taxon + score, a CRISPR hit, etc).

Line-by-line comments:

p.1: "primarily viruses, bacteria, and fungi": Archaea may deserve a mention here as well

p. 2: "encode substantial gene repertoires" may be more clear as "encode functionally diverse gene repertoires" or "encode substantial metabolic gene repertoires" ?

p. 2: "Advances in genome sequencing [...] in animal associated microbial ecosystems": I agree with this statement, but I am not sure how it is connected to the bacteriophages (the main topic of this paragraph) ?

p. 2: I believe "Viral sequencing methods influence genome assembly and community characterization" should be "Sequencing methods influence viral genome ..."

Table 1 and throughout the text: Please harmonize the use of thousand-separating commas (e.g. column "#Contigs", "15,228 protein [...] 5983 were greater").

Table 1: Please provide the length of the longest contig in kb rather than Mb.

p. 3: "PPRmeta identified the most phages, followed by [...] PPRmeta": There seems to be something wrong with this sentence. I would also note that PPRmeta is notorious for being overly "confident" in its prediction, and mistakenly predict many bacterial and especially eukaryotic genome fragments as viral. As a control, since the authors are analyzing animal-associated samples, I would encourage them to perform a simple benchmark using random fragments of honey bee genomes with the same viral sequence prediction tool, and report to the reader whether any of these eukaryotic genome fragment was predicted as viral, to better describe the potential limitations of each of the tools.

p. 3: "These numbers seemed low [...]": I agree with the authors, and it may be interesting to also test CheckV (doi: 10.1038/s41587-020-00774-7) to see if maybe a more accurate estimation of quality can be achieved ?

p. 3: "designatio." should be "designation."

p. 4: "Like Deboutte et al., we did not observe any Cystoviridae,": Since Cystoviridae harbor RNA genomes, I would not have expected these sequences to be included in the author's dataset, and I don't think any reader would ? For clarity, I would maybe suggest remove this sentence.

p. 4: Since Inoviridae and Gokushovirinae are both groups of ssDNA phages, I was not expecting any of these sequences to be included in these datasets. Can the authors speculate about why/how ssDNA phage genomes would be sequenced with their approach ? Could these be ssDNA viruses in the dsDNA prophage state (i.e. integrated into their bacterial host genome) ?

p. 5: "Microviruses predicted to infect E. coli had unknown hosts": I am confused by this sentence: were the phages predicted to infect E Coli or had no host prediction ?

p. 5: "Podoviruses [...] Myoviruses [...] Siphoviruses": These should be all lower case and not italicized (see <https://talk.ictvonline.org/information/w/faq/386/how-to-write-virus-species-and-other-taxa-names>).

p. 6: "Queuosine biosynthesis genes have been found on phage genomes and may be involved in protection from genome degradation by the host.": Please provide a reference(s) for this statement.

p. 6: "A total of 1,875 of [...] SSU rRNA": Could the authors provide some context for this number ? For instance, different metrics (including number of reads matching SSU rRNA) were recently explored across a large dataset of viromes and metagenomes (doi: 10.1038/s41587-019-0334-5) and it would be very helpful to non-specialist readers if the authors could refer to this and discuss briefly how their dataset compare to others in terms of viral enrichment.

p. 8: "phage and host abundances are well correlated": I would suggest rephrasing as "phage and host abundances can be well correlated". In my experience, there is a bias towards reporting only positive correlation, but we have a large number of examples where phage-host pairs abundance is not correlated (see e.g. 10.1038/ismej.2017.157).

p. 10: "Nearly identical contigs were collapsed using cd-hit": Was this applied only to the contig from the non-amplified PacBio library, or across multiple samples ? Cd-hit has a known limitation that it does not take into account potential "circularity" of contigs, i.e. 100% identical circular contig with a different starting position would not be recognized as 100% identical and would be put in two different clusters. If this clustering was applied to contigs from a single assembly this should not be too much of an issue as near-identical circular contigs would likely be collapsed by the assembler, but it may be worth verifying. An alternative to cd-hit in this case can be dRep, or the ani clustering provided in the CheckV package.

p. 10: Please clarify the minimum score cutoff used for host prediction with vHULK.

p. 10: For host prediction based on CRISPR, the authors should also indicated which cutoff was used on blast hits. It is often recommended to only interpret matches to CRISPR spacers with 0 or 1 mismatch over the full spacer length for host prediction (ndlr: I realized after writing this comment that this information was included in the results section p. 4, but I still believe it should be indicated here in the Methods section). It is also typically recommended to lower the word size of the blast search when dealing with CRISPR spacer (for both recommendations and further information about host prediction, the authors can refer to doi: 10.1093/femsre/fuv048 and doi: 10.1016/j.coviro.2021.05.003).

Fig. 1: "lysogenic" should probably be "temperate" here (historically, "lytic" and "lysogenic" are used to designate infection pathways, with "lytic" phages only able to undergo a lytic cycle, and "temperate" phages able to undergo either a lytic or a lysogenic cycle, see e.g. Echols, 1972, "Developmental pathways for the temperate phage: lysis vs lysogeny" <https://www.annualreviews.org/doi/pdf/10.1146/annurev.ge.06.120172.001105>).

Fig. 2: Please clarify in the legend whether sequences from RefSeq not part of a cluster including at least 1 bee microbiome phage were removed. I suspect it is the case, but for clarity it would be better if this was stated directly in the figure legend. I would also encourage the authors to highlight which sequences in the network are not from bee microbiome and instead RefSeq references (maybe with an outline ?), although this information is also somewhat provided in Fig. 5.

Supplementary Table 2: The title mentions "List of phage contigs", however some contigs have "FALSE" in the column "IsPhage" ? Please clarify

Side note: please make sure that line numbers are included in your submission

Responses to review comments are highlighted.

Comments to Editor

Dear Dr. Michela Gambino,

Thank you for reviewing our manuscript, "Global composition of the bacteriophage community in honeybees", mSystems01195-21. We appreciate the reviewer comments and have returned a much improved version of the manuscript. Below are the line-by-line responses to each reviewer.

In addition, we made a number of changes based on comments received from two experts in the field and authors of the only other two studies on honey bee bacteriophages. They both saw the biorxiv version and provided constructive feedback. Although outside of the review process for mSystems, these comments lead to improvements in the manuscript and would ideally be included in the published version. I provided a list of these changes below my responses to the mSystems reviewers.

Thank you again for publishing this important work.

Best,

James Van Leuven

Reviewer #1 (Comments for the Author):

The manuscript by Busby and colleagues investigated the bacteriophage community associated to honeybees sampled in Texas, United States, and compared these results with the ones previously obtained on two European bee populations by Bonilla-Rosso et al., 2020 and Deboutte et al., 2020. Considering that little information is available on the structure, dynamics, and function of honeybee associated viral communities, this work greatly contributes to fill in this gap. The research question is very interesting and the work done is of high quality.

The manuscript is divided in well-balanced sections. The section "introduction" describes the state-of-the-art of the study, providing to the Readers the information needed to contextualize and understand the work. The aim of the work is clearly stated at the end of the Introduction section. "Results" and "discussion" are well presented and organized. "Materials and methods" are clear and informative to allow reproduction of the experiments.

I have only few minor comments.

L. 34: please, check the presence of an extra "and".

L. 222 please check "phagess".

L. 372 and 388: please, check the upper-case letter of "melliventris".

- L. 424: Genbank accession numbers are missing.
L. 441 and 445: is "MagBio beads" the name of the kit that Authors used?
L. 453: please, check "U of I".

Thank you. All of these suggestions were made.

Reviewer #2 (Comments for the Author):

Busby et al. present the analysis of a new virome dataset from honeybee gut samples. Specifically, the authors first tested a few different approaches for sequencing the sample (including short- and long-reads, as well as with and without WGA), and then focus on the analysis of one contig collection (from non-amplified long-reads data) that is compared to two recently published honeybee virome catalogs.

Overall, I found the manuscript to be clearly written, compelling, and a clear contribution to our collective understanding of the honeybee "phageome". I only have 2 major comment that I think need to be addressed:

- The authors include a section about bacterial contamination, however I found this section to be a bit short and lacking context for a non-expert reader to interpret. I would encourage the authors to look in the literature (e.g. 10.1038/s41587-019-0334-5) to have some reference point, and discuss how the viral enrichment protocol performed in their case.

I think that it is fair to say that this section is a bit short and lacking in context for experts and non-expert readers alike. We made several changes to address this. First, we added a sentence about average read coverage of bacterial contigs vs. high-confidence viral contigs, "*Read coverages for bee-associated bacterial SSU rRNAs ranged from about 50-700X, completely within the range of phage genome read coverages*". This is helpful as a reference point for approximate abundances of phage and bacteria in our data, but does not directly address the issue of quantifying viral enrichment like a comparison between enriched and total DNA samples would. We don't have this data and cannot easily get it in time for the revision. However, we do cite recent work by Santos-Medellin et al. (2021) showing how very similar enrichment protocols improve phage detection and added the enrichment comparison paper that the reviewer suggested. Second, we added a sentence on the Bonilla-Rosso et al. (2020) paper where total DNA sequencing was done, although, not on the same sample that viral enrichment was performed: "*Bonilla-Rosso et al. found virulent phages to be much more abundant than temperate phages (90% vs. 10% of reads) in virus enriched samples, whereas, in total microbial metagenomes the pattern was opposite, suggesting that viral enrichment protocols reduce potential bacterial contamination and improve detection of phage encapsulated DNA*". Third, we performed the suggested ViromeQC analysis and included the viral enrichment estimates in the results. Details were also added to the methods section to describe how these analyses were performed. Unfortunately, there were discrepancies in the enrichment scores between our Illumina and PacBio reads (enrichment scores of 24X vs. 1.8X) and the ViromeQC paper gives no guidance about different read types. My first guess would be that the ViromeQC alignment methods work well for Illumina reads, but not well for long reads due to problems

mapping long reads to rRNA genes. In this case, the enrichment score for our Illumina data would be more accurate and a score of 24X suggests very good enrichment.

- The authors use multiple approaches for computational host prediction, which is a good idea. However, they mostly provide the consensus host prediction and not the predictions obtained with individual tools. To be able to interpret and possibly re-analyze these data, ideally the authors would provide in a supplementary table a detailed list of host prediction, with one row per "individual" prediction (e.g. vHULK taxon + score, a CRISPR hit, etc).

This information was provided in supplementary table 2, under column headers "vHulk.pred", "crisprhit", "CrisprDB", "mattInfo", "bonillaInfo", and finally, "virusHost". Descriptions of each of these column names are provided in the table legend. To this table, we added the vHulk scores under the column heading "vHulkEntropy". We added a reference to SI Table 2 in the text (line 205) where this methodology is described. Hopefully, this helps those interested in looking at the analysis more closely. We also added the full vHulk output to the github analysis page.

Line-by-line comments:

p.1: "primarily viruses, bacteria, and fungi": Archaea may deserve a mention here as well

Yes, probably so, although archaea are probably not numerically dominant? Regardless, rather than delve into whether or not to include archaea in the list or switch and use "prokaryote", we just simplified the sentence with "They". Thank you. The passage changed from "*The microbial communities in animal digestive systems are critical to host development and health. These assemblages of primarily viruses, bacteria, and fungi stimulate the immune system during development, synthesize important chemical compounds like hormones, aid in digestion, competitively exclude pathogens, etc*" to "*The microbial communities in animal digestive systems are critical to host development and health. They stimulate the immune system during development, synthesize important chemical compounds like hormones, aid in digestion, competitively exclude pathogens, etc*".

p. 2: "encode substantial gene repertoires" may be more clear as "encode functionally diverse gene repertoires" or "encode substantial metabolic gene repertoires" ?

Added "metabolic" to clarify the statement. Thank you.

p. 2: "Advances in genome sequencing [...] in animal associated microbial ecosystems": I agree with this statement, but I am not sure how it is connected to the bacteriophages (the main topic of this paragraph) ?

Yes. Thank you for pointing this out this needless statement. We merge the prior sentence and changed it from "*Phage populations in many other important ecosystems offer interesting study systems. Advances in genome sequencing throughput have invigorated an interest in our understanding of the role of microbial ecology an evolution in animal associated microbial*

ecosystems.” to “Phage populations in many other important animal associated microbial communities offer interesting study systems that will improve our understanding of the role of phages in microbial ecosystems”.

p. 2: I believe "Viral sequencing methods influence genome assembly and community characterization" should be "Sequencing methods influence viral genome ..."

Thank you. We made the suggested change.

Table 1 and throughout the text: Please harmonize the use of thousand-separating commas (e.g. column "#Contigs", "15,228 protein [...] 5983 were greater").

Thank you for pointing out this annoying inconsistency. I believe it is fixed.

Table 1: Please provide the length of the longest contig in kb rather than Mb.

We made these changes.

p. 3: "PPRmeta identified the most phages, followed by [...] PPRmeta": There seems to be something wrong with this sentence. I would also note that PPRmeta is notorious for being overly "confident" in its prediction, and mistakenly predict many bacterial and especially eukaryotic genome fragments as viral. As a control, since the authors are analyzing animal-associated samples, I would encourage them to perform a simple benchmark using random fragments of honey bee genomes with the same viral sequence prediction tool, and report to the reader whether any of these eukaryotic genome fragment was predicted as viral, to better describe the potential limitations of each of the tools.

Thank you. We fixed the error in this sentence. Our results are in agreement with the reviewers comments. PPR-meta identified over 50% more viral contigs as the next software tool (Vibrant). Worse yet, a large fraction of them were identified only by PPR-meta. We did a fairly comprehensive comparison of the tools (Supplementary Figure 1) and thus decided to require identification by at least three tools. There is an argument to be made for choosing one and carefully curating its output, but we were curious as to how they compare. To address the reviewer's comment, we did as suggested and ran PPR-meta on fragments of the bee genome. The results of this were added to line 172, *“To test PPR-Meta we compared the output from our virome assembly to computationally-generated contigs of the same length made from the A. mellifera genome. PRR-Meta identified phages in 42% of the metagenomic contigs versus only 5% in the size-matched bee genome contigs.”* The bee genome contigs and code were uploaded to Github.

p. 3: "These numbers seemed low [...]": I agree with the authors, and it may be interesting to also test CheckV (doi: 10.1038/s41587-020-00774-7) to see if maybe a more accurate estimation of quality can be achieved?

We added an analysis of the contigs using CheckV and included the results in various places throughout the manuscript (methods, results, and supplementary figures). In response to this comment, we added "*CheckV identified 3 (1%) complete, 53 (11%) high-quality, 43 (9%) medium quality, 301 (66%) low quality, and 77 (16%) undetermined viral contigs*" at line 178. These numbers are compared to Phaster and Vibrant in SI figure 2.

p. 3: "designatio." should be "designation."

Thank you. We made the suggested change.

p. 4: "Like Deboutte et al., we did not observe any Cystoviridae,": Since Cystoviridae harbor RNA genomes, I would not have expected these sequences to be included in the author's dataset, and I don't think any reader would ? For clarity, I would maybe suggest remove this sentence.

Thank you. We made the suggested change.

p. 4: Since Inoviridae and Gokushovirinae are both groups of ssDNA phages, I was not expecting any of these sequences to be included in these datasets. Can the authors speculate about why/how ssDNA phage genomes would be sequenced with their approach ? Could these be ssDNA viruses in the dsDNA prophage state (i.e. integrated into their bacterial host genome) ?

Thank you for the astute observation. It is expected that typical library prep procedures bias against ssDNA phages (see <https://doi.org/10.7717/peerj.2777>), but not completely exclude them. It is possible that we are sequencing replicative form dsDNA or perhaps as the reviewer suggests, they are prophage. Our WGA libraries are probably enriched for ssDNA phages. So, in both cases, the proportions of these viruses are likely skewed. We added a sentence to highlight this at line 202, "*We presume that these ssDNA phages were detected in an intermediate dsDNA form and thus, their abundance in the sequencing data from us and Bonilla-Rosso et al. (2020) is likely not an accurate measure of abundance of mature virions.*".

p. 5: "Microviruses predicted to infect E. coli had unknown hosts": I am confused by this sentence: were the phages predicted to infect E Coli or had no host prediction ?

Thank you for pointing out this confusing sentence. We changed "*Several other Microviruses predicted to infect E. coli had unknown hosts and ~75% blastn matches to phages from a variety of environments, but not any Microviruses used in our lab.*" to "*as well as one with ~75% blastn matches to phages from other environments. None shared sequence similarity to any of the Microvirus strains cultured in our lab. .*"

p. 5: "Podoviruses [...] Myoviruses [...] Siphoviruses": These should be all lower case and not italicized (see <https://talk.ictvonline.org/information/w/faq/386/how-to-write-virus-species-and-other-taxa-names>).

Thank you. We made the suggested changes.

p. 6: "Queuosine biosynthesis genes have been found on phage genomes and may be involved in protection from genome degradation by the host.": Please provide a reference(s) for this statement.

Thank you. We added this citation.

p. 6: "A total of 1,875 of [...] SSU rRNA": Could the authors provide some context for this number? For instance, different metrics (including number of reads matching SSU rRNA) were recently explored across a large dataset of viromes and metagenomes (doi: 10.1038/s41587-019-0334-5) and it would be very helpful to non-specialist readers if the authors could refer to this and discuss briefly how their dataset compare to others in terms of viral enrichment.

See response to comment #1, which is largely the same as this comment. We performed additional analyses and added a paragraph to the results section to address bacterial contamination. In retrospect, it would have been interesting to have sequenced a total DNA sample alongside the virome-enriched samples. However, I do not think that this is a standard control that is usually done when sequencing viromes. We think that our additional analysis provides unique and useful results, especially in the comparison between PacBio, Illumina, WGA, and non-WGA samples.

p. 8: "phage and host abundances are well correlated": I would suggest rephrasing as "phage and host abundances can be well correlated". In my experience, there is a bias towards reporting only positive correlation, but we have a large number of examples where phage-host pairs abundance is not correlated (see e.g. 10.1038/ismej.2017.157).

Thank you for the suggestion. We made this change and added the citation.

p. 10: "Nearly identical contigs were collapsed using cd-hit": Was this applied only to the contig from the non-amplified PacBio library, or across multiple samples ? Cd-hit has a known limitation that it does not take into account potential "circularity" of contigs, i.e. 100% identical circular contig with a different starting position would not be recognized as 100% identical and would be put in two different clusters. If this clustering was applied to contigs from a single assembly this should not be too much of an issue as near-identical circular contigs would likely be collapsed by the assembler, but it may be worth verifying. An alternative to cd-hit in this case can be dRep, or the ani clustering provided in the CheckV package.

Thank you for the suggestions. The clustering was performed on a single assembly of PacBio reads, which, as the reviewer points out, should collapse highly similar contigs. However, we

performed ANI clustering using CheckV and indeed found some contigs that clustered even at the 99% ANI level. Surprisingly, we found that these contigs were not circularized contigs that were split in different locations and left uncollapsed by cd-hit. In some cases these contigs were fairly different in length, in some cases they ended up in different vContact2 clusters, in some cases they had different vHulk or CRISPR predictions or had different circularity predictions. In short, we could not identify any standardized rules to apply to justify collapsing these contigs for the paper. However, many contigs with high ANI values were quite similar in length and had no descriptive information differentiating them. Thus, we collapsed some of these and provided a fasta file of what we think are a more carefully curated set of phages. We included the CheckV clustering results in SI table 2 and reported them in the results (line 220).

p. 10: Please clarify the minimum score cutoff used for host prediction with vHULK.

We used the default values for vHULK and reported the predicted host in SI table 2. We added this clarification to the methods section and added the vHULK scores to SI table 2. The entire vHULK output is now in the analysis on GitHub.

p. 10: For host prediction based on CRISPR, the authors should also indicated which cutoff was used on blast hits. It is often recommended to only interpret matches to CRISPR spacers with 0 or 1 mismatch over the full spacer length for host prediction (ndlr: I realized after writing this comment that this information was included in the results section p. 4, but I still believe it should be indicated here in the Methods section). It is also typically recommended to lower the word size of the blast search when dealing with CRISPR spacer (for both recommendations and further information about host prediction, the authors can refer to doi: 10.1093/femsre/fuv048 and doi: 10.1016/j.coviro.2021.05.003).

The exact blast commands used for searching the CRISPR databases were included in the methods sections. They read, "*blastn was used to search for Texas phage contigs in CRISPR spacer sequences from bee gut bacteria in HoneyBee-Virome-2020/pnas.2000228117.sd04.xlsx [blastn -ungapped -dust no -soft_masking false -perc_identity 100 -outfmt 6 -num_threads 4]. Second, Texas phages were compared to the DASH CRISPR database using blast (blastn -db SpacersDB.fasta -query contigs_plasmids_phage.fa -ungapped -dust no -soft_masking false -perc_identity 100 -outfmt 6 -num_threads 4)*". Results from these two searchers were reported in SI table 2.

Fig. 1: "lysogenic" should probably be "temperate" here (historically, "lytic" and "lysogenic" are used to designate infection pathways, with "lytic" phages only able to undergo a lytic cycle, and "temperate" phages able to undergo either a lytic or a lysogenic cycle, see e.g. Echols, 1972, "Developmental pathways for the temperate phage: lysis vs lysogeny" <https://www.annualreviews.org/doi/pdf/10.1146/annurev.ge.06.120172.001105>).

We went through the manuscript and changed to the temperate and virulent convention, but also parenthetically used lytic and lysogenic for the first use because the convention still is not universally used.

Fig. 2: Please clarify in the legend whether sequences from RefSeq not part of a cluster including at least 1 bee microbiome phage were removed. I suspect it is the case, but for clarity it would be better if this was stated directly in the figure legend. I would also encourage the authors to highlight which sequences in the network are not from bee microbiome and instead RefSeq references (maybe with an outline ?), although this information is also somewhat provided in Fig. 5.

Sequences from RefSeq were not included in Figure 2, only phages from the three studies on honey bees. We tried to add clarity to the legend by editing "Network diagram showing clusters of Texas honey bee phages..." to "Network diagram showing clusters of phages from honey bees...". Also, we changed a sentence in the results section to help make this more clear. "Using vContact2, we clustered the 477 Texas honey bee phages with publicly available phages (Fig 2), including VCs from Deboutte et al. and Bonilla-Rosso et al." was changed to "Using vContact2, we clustered the 477 phages from Texas honey bees with VCs from Deboutte et al. and Bonilla-Rosso et al. (Fig 2) and publicly available phage sequences (Fig 4)" at line 286.

Supplementary Table 2: The title mentions "List of phage contigs", however some contigs have "FALSE" in the column "IsPhage" ? Please clarify

Thank you for catching this. SI table 2 contains all contigs. Changed to "List of contigs".

Side note: please make sure that line numbers are included in your submission

Line numbers are included in the revised version.

Comments from two experts on the Biorxiv version

For host prediction, you used a four-step process. The order of these steps is described differently in the methods (page 10, right column third paragraph) and in the results (page 4, right column at the top). In the results, the last sentence of said paragraph implies that all these steps were done in parallel and in the case of a disagreement, you prioritised the predictions based on the order they were mentioned in the text. In the order mentioned in the results, this would mean that clustering with vCONTACT was considered with higher priority than a CRISPR match. I believe that generally, a CRISPR match is considered a better indication for a host than a clustering with a known phage.

Indeed. Thank you for pointing this out. We fixed this in the manuscript.

It is unclear from the current text how you assigned contigs to a given taxonomy. You mentioned having constructed a network with RefSeq, Deboutte's and our sequences. But then do you consider a contig to have the same taxonomy if it is clustered in the same VC? Or if it simply matches one of the sequences?

We added “*Viral taxonomy was determined based on known taxonomy of other viruses in the vContact2 clusters. Contigs with the taxonomic designation “mixed” clustered with phages from different families and the correct group could not be easily determined using blast searches to nr.*” at line 631 in the methods section.

A way to identify bacterial contamination would be to map your reads to the genomes matched by your 16S rRNA mapping, and see if the coverage for those genomes is more or less homogeneous.

As suggested by reviewers, we spent considerable effort to improve the section on bacterial contamination.

It would be super nice if you could include a table to match your clusters to our clusters, or at least to the corresponding original contig names, to facilitate the comparison between studies (I know it is a mess with the VC names, I should’ve thought of a better way to name ours for comparison). But for example so far I haven’t been able to identify which contigs correspond to your VC_106 in figure 6, which I would blast against my database locally to find them. If I could have the original contig names, that would be so much easier to identify them.

Yes, I agree. In SI table 2 we included the contig names from the two previous phage studies on honey bee phages in our vContact2 clusters. Figuring out how to find what contigs match should be fairly trivial once you know they are in this table.

PacBio sequencing still struggles with relatively high error rates of 10-15% for the CLR method you chose. How did you assess/tackle this? In the same context (and potentially answering my own question), in Table 1 you state the unamplified PacBio run has an N50 read length of 4593. In this situation, would you inevitably be in a CSS sequencing mode, i.e. sequencing most molecules multiple times and thereby reducing the error rate?

Yes, we used CSS on a PacBio Sequel II, which outputs fairly good quality reads after error correction.

Vibrant and Phaster classify the quality of most contigs as rather low (Figure S2). Do you have an idea why that might be? Could genome completeness be a factor here and would it maybe make sense to assess this e.g. with CheckV?

We added an analysis by CheckV.

In Figure 3A, you use the old names of *Lactobacillus* Firm-4 and Firm-5.

We fixed the naming convention. It should now be up-to-date.

Figure 6 shows the synteny over 13 genomes in VC_106, of which 5 are Swiss. This does not match up with Figure 5 where VC_106 (circled) should have 15-30 contigs, more than half of which Swiss. Is Figure 6 only showing a subset of VC_106?

This excellent observation is correct. We labeled the contigs in Figure 6 incorrectly (switched Swiss and Belgian colors). This error is now corrected.

According to the Venn diagram in Figure 4, of the 13 VCs shared between Switzerland, Texas and Belgium, 4 VCs should also occur in the RefSeq dataset. This means that, if I understand this correctly, in Figure 5, there should be 4 pie charts with slices representing all 4 datasets: RefSeq (black), Swiss (orange), Belgian (blue) and Texan (green). However, I can only find 2 such pies: One at the very left and one at the bottom right.

We went back and carefully checked this figure and indeed found an error. We tracked down the source of the error to an old version of our vContact2 output being loaded to make the venn diagram. While very similar, we updated our code to use the same vContact2 results as the rest of the figures.

To assess the bacterial contamination, you state in the results (page 6) and in the methods (page 11) that 3,379,211 reads were used to search for SSU rRNA matches in the SILVA database. But in the methods (page 11, second paragraph under "Sequencing") you also state that for the unamplified PacBio run, which your analyses are based on, only a total of 243,790 reads were produced.

Yes, this observation is correct. Upon review, we are not sure where the 243,790 number came from. This number is not even consistent with the numbers in Table 1. We went back to our raw data and updated the numbers in the methods section (line 581). "A total of 243,790 and 1,137,318 CLR reads were generated after demultiplexing using SMRT Link (Lima) for non-amplified and WGA samples, respectively." was changed to "A total of 3,881,928 and 11,631,485 CLR reads were generated after demultiplexing using SMRT Link (Lima) for non-amplified and WGA samples, respectively."

March 2, 2022

Dr. James T Van Leuven
University of Idaho
Department of Biological Sciences
Moscow, ID 83843

Re: mSystems01195-21R1 (Global composition of the bacteriophage community in honey bees)

Dear Dr. James T Van Leuven:

Thank you for addressing the reviewers comments. I highly appreciated that you decided to include and address also the comments from the authors of the other studies on honey bee bacteriophages. By comparing the three phage studies you did a great service to the scientific community.

Your manuscript has been accepted, and I am forwarding it to the ASM Journals Department for publication. For your reference, ASM Journals' address is given below. Before it can be scheduled for publication, your manuscript will be checked by the mSystems production staff to make sure that all elements meet the technical requirements for publication. They will contact you if anything needs to be revised before copyediting and production can begin. Otherwise, you will be notified when your proofs are ready to be viewed.

Publication Fees:

We recognize that the video files can become quite large, and so to avoid quality loss ASM suggests sending the video file via <https://www.wetransfer.com/>. When you have a final version of the video and the still ready to share, please send it to mSystems staff at mssystemsjournal@msubmit.net.

For mSystems research articles, if you would like to submit an image for consideration as the Featured Image for an issue, please contact mSystems staff at mssystemsjournal@msubmit.net.

Sincerely,

Michela Gambino
Editor, mSystems

Journals Department
Table S1: Accept
Table S5: Accept
Fig S1: Accept
Fig S2: Accept
Fig S4: Accept
Fig S5: Accept
Table S3: Accept
Table S4: Accept
Table S2: Accept
Fig S3: Accept